# A single mutation in dairy cow-associated H5N1 viruses increases receptor binding breadth

Marina R. Good [1,3], Monica L. Fernández-Quintero[2,3], Wei Ji[1,3], Alesandra J. Rodriguez[2], Julianna Han [2], Andrew B. Ward [2] & Jenna J. Guthmiller [1] ✉

Clade 2.3.4.4b H5N1 is causing an unprecedented outbreak in dairy cows in the United States. To understand if recent H5N1 viruses are changing their receptor use, we screened recombinant hemagglutinin (HA) from historical and recent 2.3.4.4b H5N1 viruses for binding to distinct glycans bearing terminal sialic acids using a glycan microarray. We find that H5 from A/Texas/37/2024, an isolate from the dairy cow outbreak, has increased binding breadth to core glycans bearing terminal α2,3 sialic acids, the avian receptor, compared to historical and recent 2.3.4.4b H5N1 viruses. We do not observe any binding to α2,6 sialic acids, the receptor used by human seasonal influenza viruses. Using molecular dynamics and a cryo-EM structure of A/Texas/37/2024 H5, we show A/Texas/37/2024 H5 is more flexible within the receptor-binding site compared to a 2.3.4.4b H5 from 2022. We identify a single mutation outside of the receptor binding site, T199I, is responsible for increased binding breadth, as it increases receptor binding site flexibility. Together, these data show recent H5N1 viruses are evolving increased receptor binding breadth which could impact the host range and cell types infected with H5N1.

Since 2021, clade 2.3.4.4b H5N1 viruses, a highly pathogenic avian influenza virus, have been causing a worldwide outbreak in wild bird populations, with reported cases on six continents. Numerous H5N1 spillover events in domestic animals, including poultry and minks, have led to massive culling events[1,2]. Moreover, H5N1 spillover into wild mammals, including aquatic and scavenger mammals, have been reported since 2022[3]. In March 2024, the United States Department of Agriculture reported an outbreak of H5N1 in domestic dairy cattle. Since then, H5N1 has expanded to 15 states with over 600 herds affected[4]. H5N1 viruses from dairy cows have spilled over into domestic felines, alpacas, poultry, and house mice[5]. Importantly, H5N1 viruses from the ongoing outbreak in dairy cattle have led to several dozen confirmed human infections, with most cases causing conjunctivitis and mild respiratory symptoms[6,7].

H5N1 infection in dairy cows is largely restricted to the mammary tissue, leading to clinical manifestations of mastitis including reductions in milk production, milk discoloration, and increased milk thickness[8]. Analysis of infectious virus revealed titers ranging from $10^4$–$10^9$ tissue culture infectious dose 50 ($TCID_{50}$) per milliliter[9,10]. One in five retail milk samples within the United States (US) has detectable viral RNA by PCR, although viable virus has not been recovered from these samples[11]. Moreover, pasteurization is an effective method to kill H5N1 viruses[9,12]. It remains unclear how H5N1 is being transmitted between cows and different hosts, although transmission is linked to raw milk consumption or exposure.

[1]Department of Immunology and Microbiology, University of Colorado Anschutz Medical Campus, Aurora, CO, USA. [2]Department of Integrative Structural and Computational Biology, The Scripps Research Institute, La Jolla, CA, USA. [3]These authors contributed equally: Marina R. Good, Monica L. Fernández-Quintero, Wei Ji. ✉e-mail: jenna.guthmiller@cuanschutz.edu

Avian influenza viruses, including H5N1, preferentially bind glycans bearing terminal α2,3 sialic acids[13]. In contrast, influenza viruses that cause seasonal influenza outbreaks in humans prefer glycans bearing terminal α2,6 sialic acids[14]. The influenza virus preference for α2,3 or α2,6 sialic acid linkages creates a major species barrier for avian influenza viruses to spill over into humans. Two recent studies show that dairy cow mammary tissue, and particularly the mammary alveoli, has abundant α2,3 sialic acid-linked glycans[15,16]. Moreover, dairy cow mammary tissues also have α2,6 sialic acid-linked glycans[15,16], albeit whether these α2,6 sialic acid-linked glycans can mediate viral entry remains unclear.

In this study, we investigated if recent H5N1 viruses are evolving their receptor binding specificities. We identified that H5 from the ongoing dairy cow outbreak has increased binding breadth to backbone glycans bearing α2,3 sialic acids relative to other H5N1 viruses, which was linked to a single mutation near, but not within, the receptor binding site (RBS). I199 emerged in late 2023, before the onset of the ongoing dairy cow outbreak, and is now the dominant amino acid at this residue in North American isolates. Our study

indicates a single mutation near the RBS expands the types of backbone glycans bound by H5, which could imply an increase in cell, tissue, and host tropism.

## Results

### HA from circulating H5N1 in dairy cows is phylogenetically distinct

The phylogenetic analysis of HA genetic sequences from 94 H5Nx viruses representing various clades and ancestral and human H5N1 sequences revealed distinct branching patterns (Fig. 1; Supplementary Data 1). Human H5N1 sequences from Vietnam (2004) and Indonesia (2005) were closely related but formed separate branches from the 2.3.4.4 clusters, bridging the evolutionary paths between the ancestral A/Goose/Guangdong/1996 sequence and the 2.3.4.4 clades (Fig. 1). The 2.3.4.4 clade is further subdivided into distinct subclusters (2.3.4.4b, 2.3.4.4c, 2.3.4.4e, 2.3.4.4 g, 2.3.4.4 h), highlighting the diversity and evolutionary progression of H5Nx across various regions. The 2.3.4.4b clade represents the dominant H5N1 viruses globally since 2021[17–19]. 2.3.4.4.b H5N1 viruses segregated based on continent(s), with

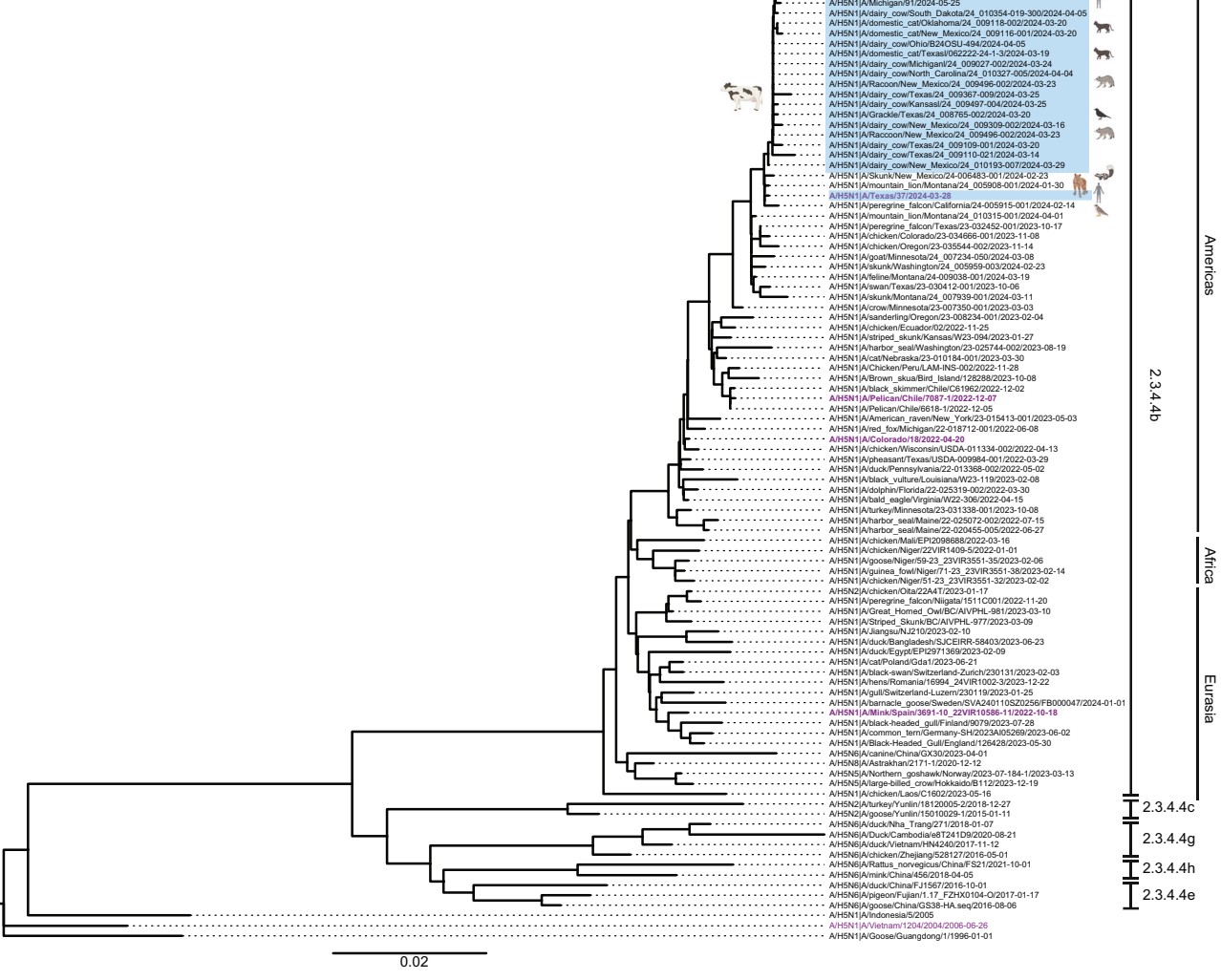

**Fig. 1 | Phylogenetic tree of highly pathogenic avian H5N1.** The Neighbor-Joining (NJ) phylogenetic analysis of 94 hemagglutinin gene sequences. Clade 2.3.4.4 is subdivided into distinct subclades, including 2.3.4.4e, 2.3.4.4 h, 2.3.4.4 g, 2.3.4.4c, and the currently dominant 2.3.4.4b clade. The new group belonging to clade 2.3.4.4b includes strains isolated from domestic dairy cows and humans and animals linked to H5N1-positive dairy farms (highlighted with a blue region and animal symbols) in the United States in 2024. Distinct clades of the virus are labeled on the

right side of the figure. Tips are labeled with H5Nx strain names, host species, and isolation dates. Nodes represent inferred common ancestors of the grouped tips. Branch lengths are proportional to the number of nucleotide substitutions per site, indicating the divergence between nodes. Sequences used for analysis are in Supplementary Data 1. Symbols depicting various host animals was in part created in BioRender. Ji, W. (2024) https://BioRender.com/f7Ol446.

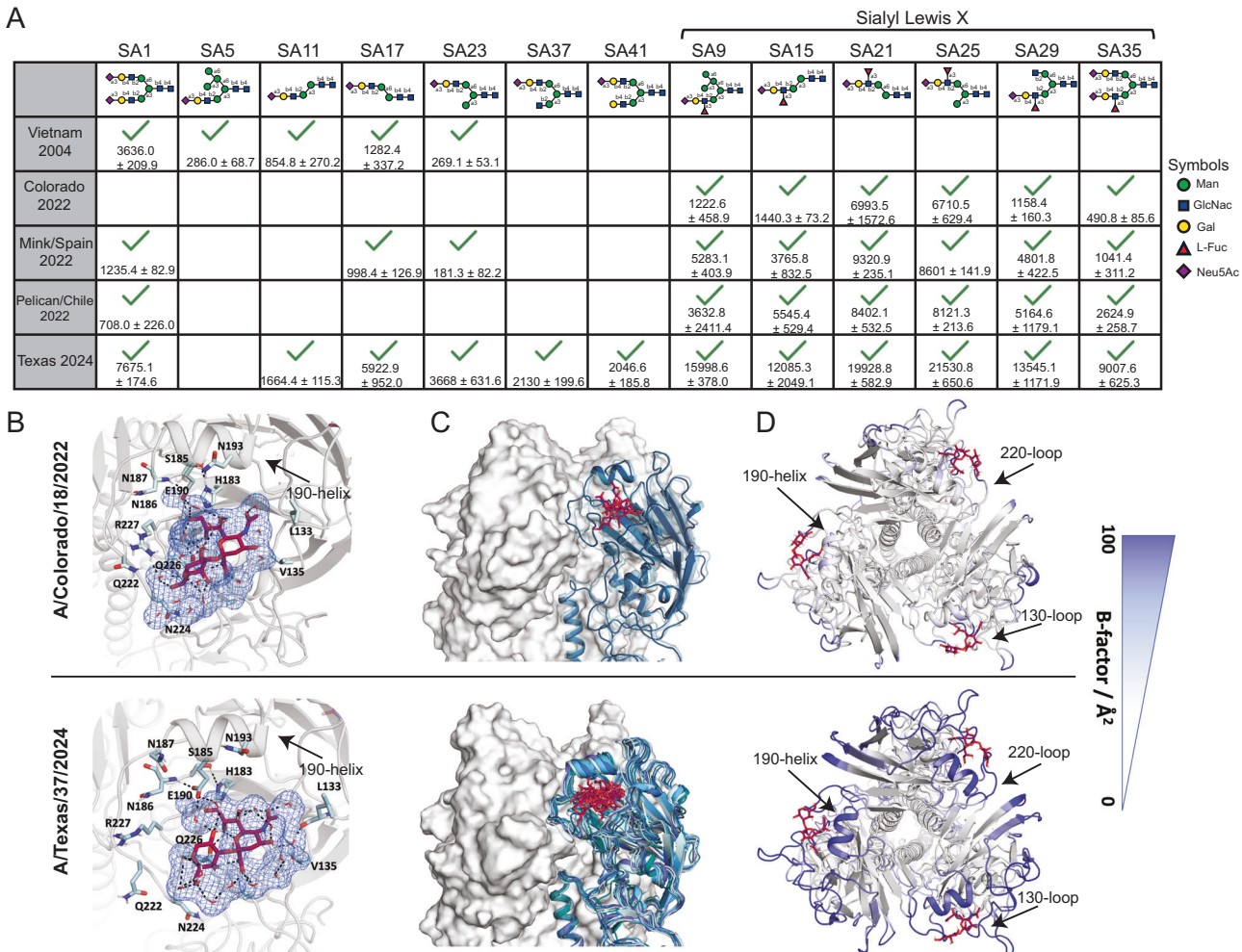

**Fig. 2 | A dairy cow associated H5N1 virus exhibits increased glycan binding breadth. A** rH5 binding to distinct Neu5Ac glycans. Green checkmarks indicate a positive binding result for the corresponding glycan. Normalized relative fluorescence unit (RFU) values ± standard deviation are indicated below checkmarks. Value above each glycan indicates the glycan number in the array. Symbols represent mannose (man), N-acetylglucosamine (GlcNAc), galactose (Gal), L-Fucose (L-Fuc), and N-acetylneuraminic acid (Neu5Ac). Glycans studied are in Supplementary Data 2. Source data are provided in the source data file. **B−D** MD simulations of A/Colorado/18/2022 and A/Texas/37/2024 to characterize the LSTa binding site properties. **B** Representative structure obtained from MD simulations showing interactions of LSTa (burgundy) with A/Colorado/18/2022 and A/Texas/37/2024 in solution. **C** Conformational states of A/Colorado/18/2022 and A/Texas/37/2024 binding to LSTa. Each shade of blue represents a distinct conformation. Molecular coordinates are provided in Supplementary Data 3. **D** Residue-wise B-factor, as a measure of flexibility, mapped on the respective A/Colorado/18/2022 and A/Texas/37/2024 structure.

virus isolates from Eurasia and Africa more closely related to each other than to viruses from North and South America, which was independent of isolation date (Fig. 1). Moreover, viruses from North and South America are more closely related to each other than they are to viruses from Eurasia and Africa (Fig. 1). Within the Americas branches, the cattle-derived H5N1 viruses form a distinct group within the 2.3.4.4b clade (Fig. 1). The new group shows a cluster of H5N1 strains isolated from various hosts in the United States in 2024, including dairy cows, domestic cats, raccoons, skunks, mountain lions, and several bird species. The clustering of sequences from the recent dairy cow outbreak indicates a common ancestor and suggests a potential transmission link between these species. Together, these data demonstrate that the recent outbreak of H5N1 in dairy cows is distinct from other circulating 2.3.4.4b H5N1 viruses.

### Dairy cow-related H5 has increased glycan binding breadth
To understand if recent H5 has changed its receptor binding specificity, we tested recombinant H5 (rH5) from an ancestral H5N1 virus, 2.3.4.4b H5N1 viruses from 2022, and a recent H5 isolated from dairy farm worker (A/Texas/37/2024) on a N-acetylneuraminic (Neu5Ac) and

N-glyconeuraminic acid (Neu5Gc) glycan microarray (Supplementary Data 2). This microarray includes an array of distinct glycans with terminal sialic acids of both the α2,3 and α2,6 Neu5Ac linkages, which correspond to the receptors for avian and human influenza viruses, respectively. This array also includes α2,3 and α2,6 Neu5Gc linkages, which are not expressed in humans or most birds[20,21]. Notably, we did not observe any binding of HA to glycans bearing Neu5Gc (Supplementary Fig. 1). This microarray includes Neu5Ac glycans with distinct branches, with most glycans incorporating an α2,3 or α2,6 Neu5Ac linkage onto a single branch of a single N-acetyllactosamine or sialyl Lewis X moiety (α2,3 only) and a second non-sialylated branch of varying compositions (Supplementary Data 2). Previous studies using this same microarray have shown specific binding of H1N1 to α2,6 linked sialic acids[22,23]. We observed ancestral rH5 from A/Vietnam/ 1204/2004 exhibited a dominant preference for glycans bearing α2,3 linked to an N-acetyllactosamine (Fig. 2A). In contrast, rH5 from A/ Colorado/18/2022, which was isolated from a human involved in culling 2.3.4.4b H5N1 infected poultry, exhibited restricted binding to 3′ sialyl Lewis X glycans. Other rH5 from 2022 2.3.4.4b H5N1 viruses revealed expanded binding breadth to 3′ sialyl Lewis X and α2,3 linked

N-acetyllactosamine glycans (Fig. 2A). Compared to other 2.3.4.4b rH5s, A/Texas/37/2024 has gained further binding breadth to nearly all α2,3 sialic linked N-acetyllactosamine glycans, including those with asymmetrical branches (Fig. 2A). Moreover, A/Texas/37/2024 had augmented binding signal for 3′ sialyl Lewis X glycans relative other 2.3.4.4b H5 viruses and to α2,3 sialic acid-linked N-acetyllactosamine glycans relative to A/Vietnam/1204/2004 (Fig. 2A; Supplementary Fig. 1A–E). Importantly, we did not observe any binding to glycans bearing only terminal α2,6 sialic acids, indicating recent H5N1 viruses have not gained binding affinity to receptors used by human seasonal influenza virus subtypes (Supplementary Fig. 1A–E).

To understand the mechanism of how A/Texas/37/2024 rH5 has gained increased binding breadth, we performed molecular dynamics (MD) simulations. Sequences of rH5 from A/Texas/37/2024 and A/Colorado/18/2022 were used to model binding to LSTa, an α2,3 sialic acid avian analog receptor (Fig. 2B, Supplementary Data 3). We identified similar residues of both A/Colorado/18/2022 and A/Texas/37/2024 involved in LSTa binding (Fig. 2B). Additionally, we observed a strong water-mediated, hydrogen bond network of LSTa with both A/Texas/37/2024 and A/Colorado/18/2022. In particular, we found a long-lasting water-mediated hydrogen bond of Sia-1 (O8) of LSTa with the sidechain carboxyl of residue E190 (Fig. 2B, Supplementary Fig. 2), an important residue for mediating α2,3 sialic acid receptor specificity[24]. A/Texas/37/2024 exhibited more variability in the receptor binding site (RBS), adopting 9 different conformations (Fig. 2C). In contrast A/Colorado/18/2022 reveals a more stable RBS with only 3 conformational states (Fig. 2C). To quantify the differences in flexibility, we mapped the B-factor obtained from the MD simulations onto the structure of A/Colorado/18/2022 and A/Texas/372/2024 in complex with LSTa. A/Texas/37/2024 exhibited overall a dramatically higher B-factor. This is particularly pronounced in the 190-helix, whereas A/Colorado/18/2022 revealed an overall low B-factor (Fig. 2D). Thus, our findings suggest that the expanded binding breadth to glycans even bearing terminal α2,3 sialic acids, is mediated by an increased flexibility within the RBS of A/Texas/37/2024.

### Cryo-EM structure of A/Texas/37/2024 HA in complex with LSTa

To ensure the reliability of our predictions, we determined the cryo-EM structure of A/Texas/37/2024 in complex with LSTa and broadly neutralizing monoclonal antibody CR9114 (Fig. 3A; Supplementary Fig. 3; Supplementary Table 1). Overlay of the modeled structure and the cryo-EM structure of A/Texas/37/2024 revealed a nearly identical structure and positioning of LSTa within the RBS (Supplementary Fig. 3E). A/Texas/37/2024 binding to LSTa based on the experimentally determined structure highlights the critical role hydrogen bonding of the sidechains of residues H183, S185, N186, E190, and Q226 for LSTa binding (Fig. 3B, C; Supplementary Fig. 3). Moreover, Van der Waals interactions with L133 and Sia-C11 and backbone hydrogen bonding between V135 and R225 with LSTa further support HA binding to LSTa (Fig. 3B, C). The binding mode of CR9114 in complex with A/Texas/37/2024 is almost identical to CR9114 binding to A/Vietnam/1203/2004 H5 crystal structure (PDB: 4FQI)[25], highlighting the central stalk epitope is highly conserved among A/Texas/37/2024 and A/Vietnam/1203/2004 (Supplementary Fig. 3F).

### H5N1 viruses circulating in the Americas have acquired mutations near the RBS

Several mutations have arisen in 2.3.4.4b viruses since 2022, particularly at L111M, T199I, and V214A (Fig. 4A). Notably, these three mutations lie outside of the traditional RBS, which is comprised of the 130-loop, 190-helix, and 220-loop[26]. Analysis of these mutations based on location and outbreak revealed that all three mutations are specific to H5N1 viruses in the Americas (Fig. 4B). Importantly, we observed T199I was found in all H5N1 viruses from the dairy cow outbreak, as well as some H5N1 viruses circulating in the Americas not related to the

ongoing dairy cow outbreak (Fig. 4B). T199I is the only amino acid difference between A/Texas/37/2024 and A/pelican/Chile/7087-1/2022 (Fig. 4C, D). Importantly, HAs from the ongoing dairy cow outbreak are highly conserved, as the amino acid sequence of A/Texas/37/2024 and A/Michigan/90/2024 are identical and the nucleotide sequence of HA1 has three synonymous mutation differences (Fig. 4C; Supplementary Fig. 4). Structurally, position 199 is located on the backside of the 190-helix (Fig. 4A). We observed that the T199I mutation arose in the second half of 2023, with I199 becoming dominate by November 2023 (Fig. 4D, E). Notably, A/Texas/37/2024 HA NT sequence clusters more closely with an HA sequence collected from a mountain lion, also known as a cougar (*Puma concolor*) in Montana than sequences related to the ongoing dairy outbreak (Fig. 4E). Interestingly, the mountain lion isolate was collected in January 2024 and it was recently proposed that the dairy cow outbreak has been ongoing since late 2023[27]. These data would suggest a closely related common ancestor from the mountain lion case to the ongoing dairy cow outbreak. Together, these data show that H5N1 viruses in the Americas have accumulated mutations within the RBS, with T199I being the only mutation specific to the ongoing dairy cow outbreak.

### T199I is responsible for increased α2,3 sialic acid binding breadth

T199I resides in a loop on the backside of the 190-helix that leads into the 220-loop of the RBS. To determine if T199I augments binding breadth, we reverted A/Texas/37/2024 from I199 to T199 (I199T) and tested glycan binding breadth. A/Texas/37/2024 I199T demonstrated identical binding breadth to A/pelican/Chile/7087-1/2022 (Fig. 5A, B; Supplementary Fig. 1F), indicating a mutation outside of the RBS massively affects receptor binding specificity. Furthermore, we find that T199 hydroxyl group hydrogen bonds directly or via water-bridges with the sidechain of residue N197 (~65% of the simulation time) and N248 (~80% of the simulation time) on the same protomer and with R212 on the same protomer (~30% of the simulation time) and with R212 on the neighboring protomer (~30% of the simulation time; Fig. 5C, Supplementary Data 3). Thus, the T199I mutation would lose these stabilizing hydrogen bonds, leading to more flexibility within the RBS. This additional stabilization of T199I is further emphasized by a more favorable interaction energy of T199 compared to I199 (~−60 kcal/mol to ~−45 kcal/mol). Analysis of the B-factor of A/Texas/37/2024 with T199 shows a decreased flexibility within the RBS, leading to 130 and 220 loops having similar flexibility to A/Colorado/18/2022, and distinct from A/Texas/37/2024 with the naturally occurring I199 (Fig. 5D, E). These data demonstrate that a single mutation outside the RBS can improve binding breadth to distinct backbone glycans bearing terminal α2,3 sialic acids. Mechanistically, we propose that T199 stabilizes the RBS, leading to more restricted receptor binding.

## Discussion

Our study shows that H5N1 viruses from the ongoing dairy cow outbreak have increased their receptor binding breadth to bind more glycans bearing α2,3 sialic acids. We observed that A/Texas/37/2024 could bind 3′ sialyl Lewis X glycans, which has a fucosylated sialoside, and α2,3 sialic acid-linked N-acetyllactosamine glycans. We observed a historical H5N1 virus, A/Vietnam/1204/2004, preferentially bound to α2,3 sialic acid-linked N-acetyllactosamine glycans, whereas A/Colorado/18/2022, the first human case of 2.3.4.4b virus in the US, was highly specific to glycan with a 3′ sialyl Lewis X structure. A prior study found A/Vietnam/1194/2004, an isolate closely related to A/Vietnam/1204/2004, binds both 2,3 sialic acid-linked N-acetyllactosamine glycans and 3′ sialyl Lewis X, albeit the former with 3-times stronger affinity[28]. Moreover, an analysis showed Asian 2003–2004 H5 isolates from chickens and humans preferred sulfated α2,3 sialic acid-linked glycans, including binding to a sulfated sialyl Lewis X glycan[13]. Avian influenza viruses are postulated to have restricted sialic acid binding breadth as

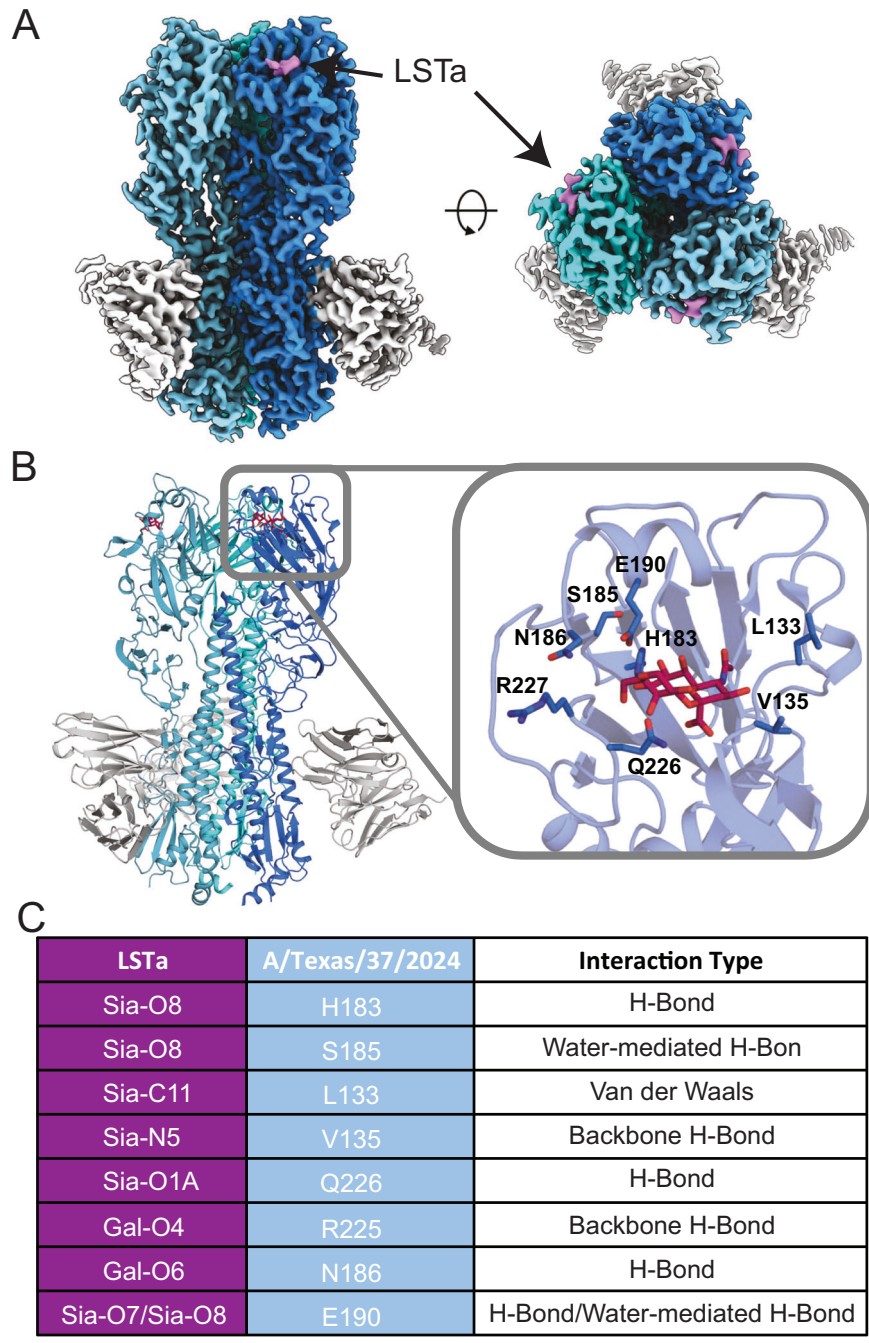

**Fig. 3 | Cryo-EM structure of A/Texas/37/2024 HA in complex with LSTa and CR9114. A** Cryo-EM structure of rH5 from A/Texas/37/2024 bound with the sialic acid and first galactose-2 of LSTa (magenta) and CR9114 (gray). Each HA protomer is a distinct shade of blue. **B** Ribbon structure of rH5 from A/Texas/37/2024 highlighting the major H5 contacts with LSTa (maroon). **C** HA contacts and interactions with LSTa.

a mechanism to have specific and limited host tropism[29]. As it stands, our understanding of the glycan structures bearing α2,3 sialic acids on distinct cell types, tissues, and hosts remains poorly understood. Moreover, how host glycosylation patterns impact influenza virus evolution to augment receptor binding affinity and breadth is not well characterized. Importantly, it was observed that cow mammary tissues have both Neu5Ac and Neu5Gc, with Neu5Ac 10–100 times more abundant in milk than Neu5Gc[30,31]. It is likely that α2,3 Neu5Ac glycans are the receptors for the outbreak of H5N1 in dairy cows as our study highlights dairy cow-associated H5 has high specificity for α2,3 Neu5Ac

and recent studies have shown H5 can strongly bind to cow mammary tissues[16] and direct inoculation of cow udders leads to robust infection[32]. Thus, a deeper understanding of how glycan binding specificity and breadth across diverse hosts is needed to perform risk assessment of potential pandemic influenza viruses, such as H5Nx.

2.3.4.4 viruses in the mid-2010s gained mutations at positions K222Q and S227R, which increase binding to fucosylated sialosides, such as 3' sialyl Lewis X[33]. K222 sterically clashes with the fucose group on 3' sialyl Lewis X[33], which could explain the selection of mutations at this site that improves binding to fucosylated glycans. Importantly,

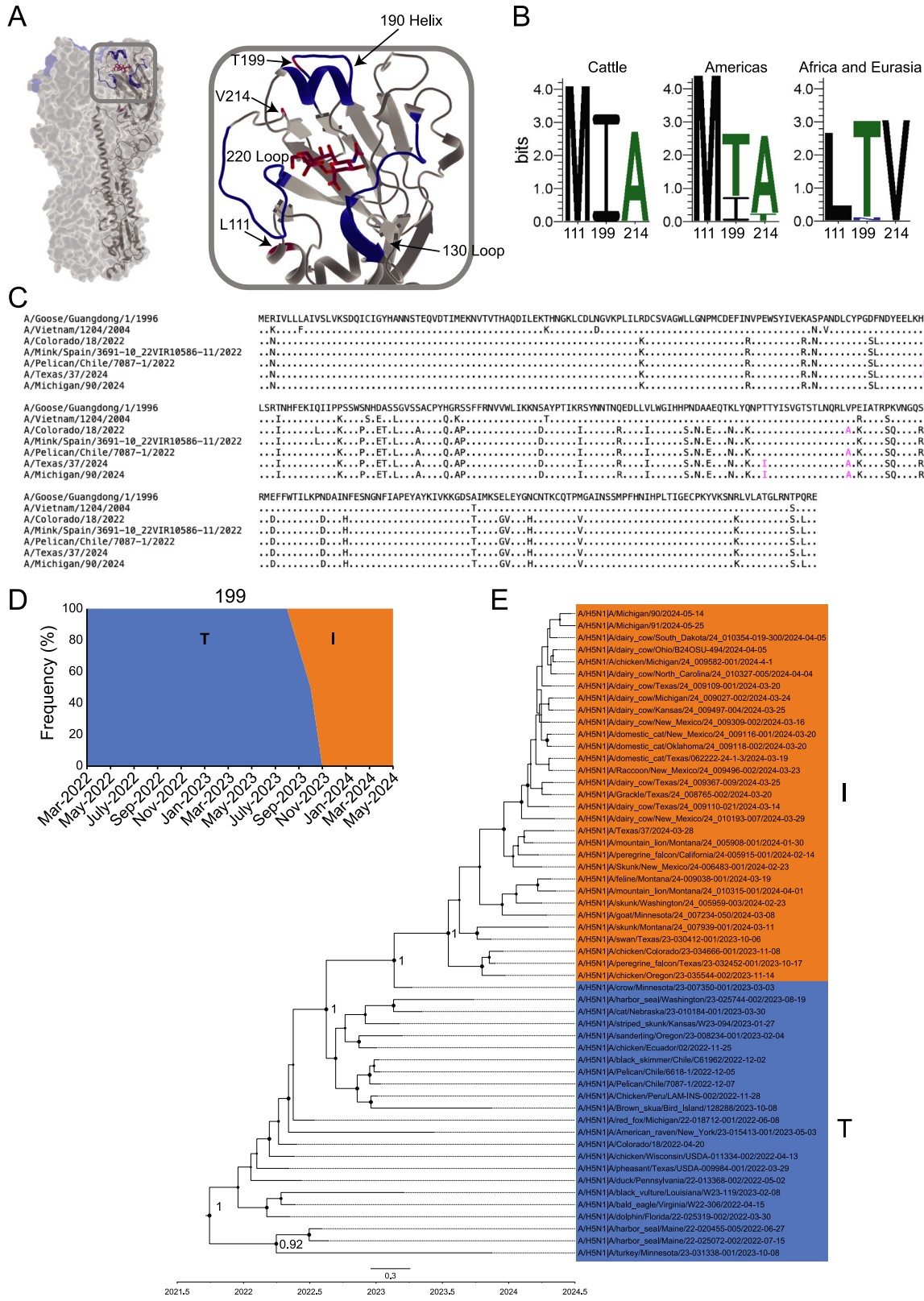

2.3.4.4b clade H5N1 viruses have retained K222Q and S227R, which could help explain their preferential binding to 3' sialyl Lewis X. Our data adds T199I to the list of mutations that change receptor binding, by expanding the H5 RBS binding to α2,3 sialic acid-linked N-acetyllactosamine glycans. Our MD data shows that T199 stabilizes the RBS through hydrogen bonds formed with N248 on the same protomer. Since T199 is within a loop directly following the 190-helix and leads

into the 220-loop, these hydrogen bonds likely stabilize the RBS, limiting the number of conformations it can adopt while binding α2,3 sialic acid-linked glycans.

Our glycan binding data shows that 2.3.4.4b H5N1 viruses, including those related to the ongoing dairy cow outbreak, have not gained binding to α2,6 sialic acids, the most abundant human receptor for influenza viruses. A recently published study by Eisfeld et al.

**Fig. 4 | 2.3.4.4b H5N1 viruses in the Americas recently acquired T199I.**
**A** Structural depiction on A/Texas/37/2024 of the RBS and recent mutations. Blue residues indicate those found within the 130-loop, 190-helix, or 220-loop. Red residues indicate mutations of interest. **B** Logo plots of positions 111, 199, and 214 based on geographical location. The Americas logo plot does not include sequences from the dairy cow outbreak. **C** Amino acid alignment of HA1 from H5N1 viruses in this study. Residues in magenta are positions, 111, 199, and 214.
**D** Frequency of T199 (blue) and I199 (orange) in circulating 2.3.4.4b H5N1 viruses in

the Americas, including the dairy cow outbreak, between March 2022 and May 2024. Source data are provided in the source data file. **E** Maximum Clade Credibility tree of 2.3.4.4b clade H5N1 viruses in the Americas with T199 or I199 from 2022 to 2024. Posterior probabilities were marked with black dots at the nodes, with the main clades labeled by number. The size of the dots corresponds to the posterior probability values. The larger the black dot, the higher the value it represents. The scale bar at the bottom represents 0.3 substitutions per site.

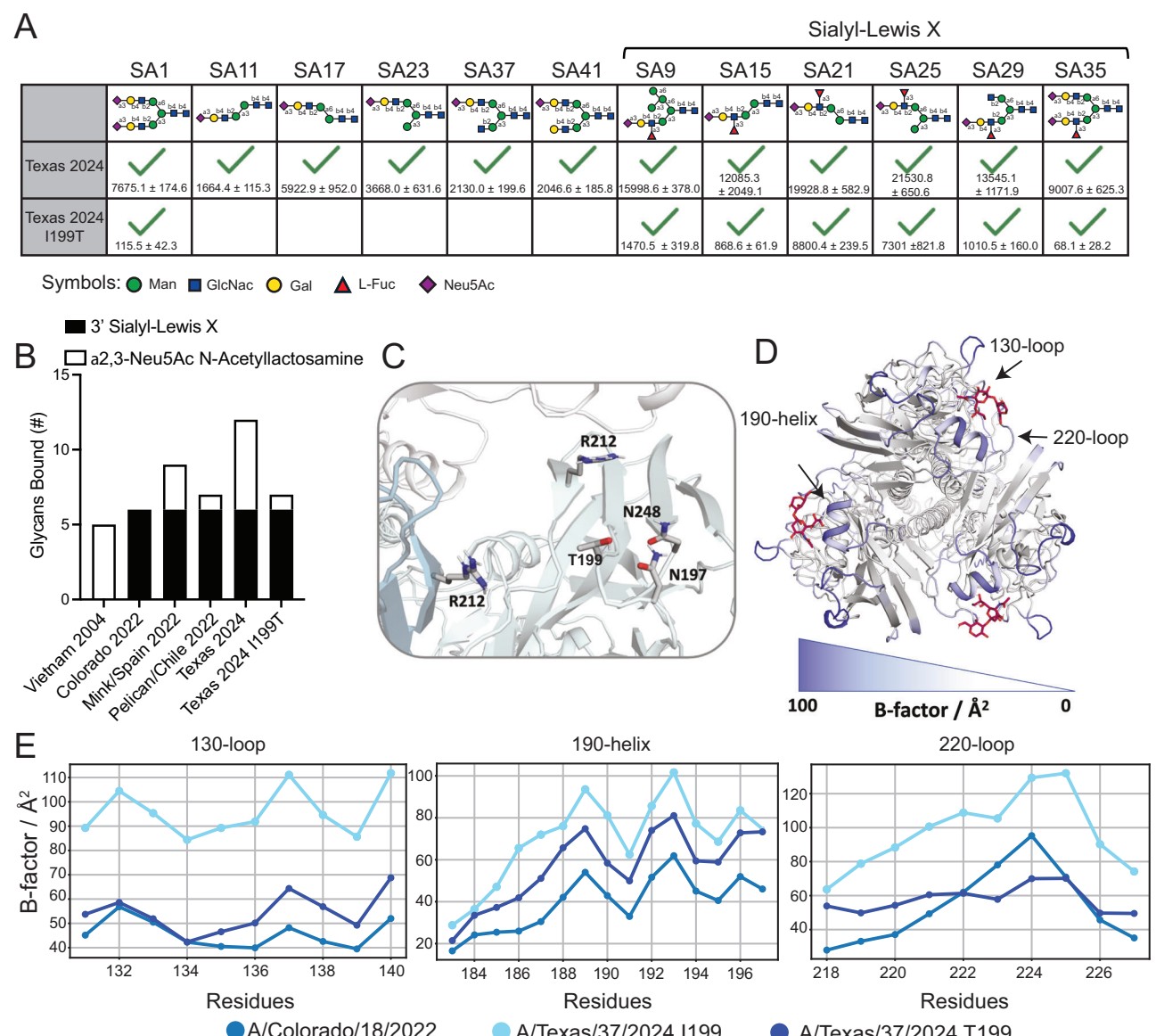

**Fig. 5 | T199I is responsible for increased glycan binding breadth in A/Texas/37/2024. A** Glycan binding profiles of WT A/Texas/37/2024 and the I199T mutant of A/Texas/37/2024 to distinct Neu5Ac glycans. Green checkmarks indicate a positive binding result for the corresponding glycan. Normalized relative fluorescence unit (RFU) values ± standard deviation are indicated below checkmarks. Only glycans above the background signal are shown. Symbols represent mannose (man), N-acetylglucosamine (GlcNAc), galactose (Gal), L-Fucose (L-Fuc), and

N-acetylneuraminic acid (Neu5Ac). **B** Number and type of glycans bound by each H5. **C** MD model of residues involved in direct or water mediated hydrogen bonds with T199 in the MD simulation. Molecular coordinates are provided in Supplementary Data 3. **D** B-factor analysis of A/Texas/37/2024 I199T. **E** Summary of individual residues' B-factor for the 130- and 220-loops and the 190-helix of A/Colorado/18/2022, A/Texas/37/2024, and the I199T mutant of A/Texas/37/2024 H5. Source data are provided in the source data file.

showed binding of A/dairy cattle/New Mexico/A240920343-93/2024 H5N1 virus to both α2,3 and α2,6 linked sialic acids, which contradicts our study[34]. The strain used in this study possesses only one amino acid difference in HA1 (S323N) in a site distal from the RBS and, therefore, is unlikely to affect receptor binding. Moreover, two preprints in response to the Eisfeld et al. study are consistent with our findings that

dairy cow-associated H5 is highly specific to α2,3 linked sialic acids[35,36]. As these two preprints use multiple complementary approaches to define receptor specificity using slightly different strains, it is likely that H5N1 viruses in dairy cows have retained specificity to α2,3 Neu5Ac glycans. Moreover, dairy cow associated H5N1 viruses have not acquired any known mutations that would lead to a receptor switch.

Two mutations, E190D and G225D, are defined mutations for receptor switches between α2,3 and α2,6 sialic acid-linked glycans[24]. E190 and D190 function as direct α2,3 and α2,6 sialic acid contacts, respectively[24,37], whereas G225D mutation introduces a bulky amino acid within the 220-loop, making binding specific to α2,6 sialic acids[38,39]. While mutations at these two sites are not observed within the circulating 2.3.4.4b, our data support that a mutation near the RBS is impacting receptor binding specificity and breadth to glycans bearing α2,3 sialic acids. Notably, our data supports that mutations not directly within the RBS can dramatically change receptor binding properties. Deep mutational scanning tools for emerging influenza viruses can provide insight into mutations permissible for increased binding to α2,3 and α2,6 sialic acids[40]. A proactive analysis of H5 sequences and their potential to increase binding breadth and specificity can alert to their potential to cause a pandemic.

## Study limitations

A limitation of our study is that we only used recombinant HA to study HA-glycan interactions, which will be limited to low avidity interactions. As a result, our approach is only detecting high affinity interactions but not low affinity high avidity interactions that would be detected using a viral particle. The glycan microarray did not include sulfated glycans, which are known to be recognized by H5Nx viruses[13]. Sulfated glycans with Neu5Ac may be an important glycan specificity of recent H5N1 viruses. Moreover, the glycan microarray only includes a single N-acetyllactosamine and do not have poly-N-acetyllactosamine with terminal sialic acids. Therefore, a caveat of our study is that we did not test HA binding sialylated poly-N-acetyllactosamines, which are known to be the preferred receptors for recent H3N2 viruses[41]. Lastly, our study used LSTa to model H5-sialic acid binding. LSTa is not a physiologically relevant glycan structure for understanding host glycan binding. However, LSTa is a useful model sialoside for study interactions between HA RBS and sialic acid. Future studies will focus on using more physiologically relevant glycan structures to understand glycan binding properties.

## Methods

### Sequence analyses

We downloaded 94 H5Nx sequences (Supplementary Data 1) from different clades (2.3.4.4b, 2.3.4.4c, 2.3.4.4e, 2.3.4.4g, 2.3.4.4h) along with the ancestral H5N1 sequence (A/Goose/Guangdong/1996-01-01), and two human H5N1 sequences from Asia (A/Vietnam/2004 and A/Indonesia/2005) from the GISAID database[42–44]. H5N1 avian influenza virus sequences from the 2.3.4.4b clade in the Americas were aligned using MEGA11[45,46]. The best-fit nucleotide substitution model was identified using MEGA11. A Maximum Clade Credibility (MCC) tree was constructed with BEAST v2.6.3, using the TN93+Gamma5 substitution model and partitioning by positions 1, 2, and 3[47]. The analysis employed an uncorrelated relaxed clock with a chain length of 10,000,000 generations, sampling every 1000 generations, and discarding 10% of the samples as burn-in. The resulting file was analyzed and annotated using Tracer v1.7.1 and TreeAnnotator v1.10.4, and the annotated MCC tree was visualized using FigTree v1.4.4[48,49]. The mammal symbol on the tree originates from the Pixabay website and BioRender. Weblogo plots were generated using the WebLogo generator[50].

### Cloning and protein purification

HA sequences were downloaded from GISAID. HA ectodomains were synthesized from Integrated DNA Technologies (IDT) or Twist Biosciences and cloned into a vector with a Fibritin Foldon Domain and a his-tag. PCR-based site-directed mutagenesis was used to introduce I199T into the A/Texas/37/2024 construct. PCR reactions with mutagenesis primers were performed PrimeSTAR Max DNA Polymerase (Takara). The PCR product was treated with DpnI (New England Biolabs). All plasmids were transformed into *E. coli* New England Biolabs),

miniprepped, correct clones selected, and maxiprepped. Sequence verified maxipreps were used for transfections in HEK293T cells (ATCC) or Expi293F Cells (Thermofisher). Expi293F suspension cells were maintained at 125 rpm at 37 °C with 8% $CO_2$ in FreeStyle™293 expression medium (Gibco). HEK293T cells were grown in 37 °C with 5% $CO_2$. HAs and CR9114[25] were produced in-house via transfection of HEK293T cells or Expi293F cells. HAs were purified from the supernatant using nickel-NTA agarose (Qiagen) and disposable 10 mL polypropylene columns (Thermofisher). HA concentrations were determined using a Pierce BCA Protein Assay (Thermofisher). HA was aliquoted and maintained at −80 °C.

### Neu5Ac and Neu5Gc glycan microarray

We used a Neu5Ac and Neu5Gc glycan microarray (Zbiotech; Lot # 04242301); structures can be found here (https://www.zbiotech.com/product/neu5gc-neu5ac-n-glycan-microarray/). HA proteins were diluted into 200 μL of glycan microarray assay buffer (GAAB) supplemented with 1% BSA, for a final concentration 40, 20 or 10 μg/mL of HA. Subsequently, an anti-6x His tag antibody and anti-rabbit immunoglobulin (H + L) (Cy3) antibody were added to the HA + GAAB mixture at a final concentration of 3.2 μg/mL each. The entire mixture was then incubated at room temperature for 60 min with gentle vortexing as part of the precomplexing process. To prepare the microarray slide for analysis, the slide was pretreated with glycan microarray blocking buffer (GABB) supplemented with 1% BSA at room temperature for 60 min. Following this, the precomplexed HA protein samples were added to the microarray, with 100 μL added to each submicroarray. The slide was incubated for 60 min at room temperature to facilitate binding interactions. After this incubation period, the slide was thoroughly washed to remove any unbound components. The slide was then scanned at 532 nm using high intensity (1 PMT) to detect and visualize any interactions. Innopsys' Mapix software was used to analyze the microarray scans. Positive binding signals were determined by subtracting the background and negative control signals from all experimental sample signals.

### MD simulations

The starting structures of A/Colorado/18/2022, A/Texas/37/2024, and A/Texas/37/2024 I199T, were predicted using *ColabFold*, which increased the accessibility of protein structure prediction tools by combining *AF2* with the rapid homology search capability of MMseqs2, making it an easy to use and fast software (-90-fold speed up in prediction) to predict homo-and heteromeric complexes, matching the prediction quality of AF2 and AF-multimer[51,52]. The reliability and quality of the predicted structural models were ensured by a PLDDT score exceeding 90%. As template to model the sialic acid complex (LSTa) we used the available X-ray structure of A/California/04/2009 in complex with LSTa (PDB ID: 3UBJ). In addition, we used the GlycoShape tool to ensure the known glycosylation sites are glycosylated for the simulations and compared the predicted glycosylation sites with available structures[53]. For our simulations we capped the C-terminal and N-terminal parts of each domain with acetylamide and N-methylamide to avoid perturbations by free charged functional groups. For each H5 variant, we performed three repetitions of 500 ns of classical molecular dynamics simulations using the AMBER 22 simulation software package which contains the pmemd.cuda module[54]. The structures were prepared using CHARMM-GUI[55,56]. The structure models were placed into cubic water boxes of TIP3P water molecules[57] with a minimum wall distance to the protein of 12 Å and the charge was neutralized with $K^+$ $Cl^-$ ions up to a concentration of 0.15 mM[58,59]. Parameters for all simulations were derived from the AMBER force field 14SB[60,61]. Each system was carefully equilibrated using a multistep equilibration protocol[62]. Bonds involving hydrogen atoms were restrained using the SHAKE algorithm, allowing a timestep of 2.0 femtoseconds[63]. The systems' pressure was maintained at 1 bar

by applying weak coupling to an external bath using the Berendsen algorithm[64]. The Langevin Thermostat was utilized to keep the temperature at 300 K during the simulations[65].

## MD analysis

For all investigated H5 variants, we calculated the respective contacts of the HA protomers with LSTa in solution using the GetContacts software (Stanford University; https://getcontacts.github.io/). This tool can compute interactions within one protein structure, but also between different protein interfaces and allows to monitor the evolution of contacts during the simulation. An interaction/water-mediate interaction was defined based on a 3.5 Å heavy atom distance-cutoff criteria. We added these details to the methods. Apart from visualizing and quantifying the contacts of the different poses, we calculated the residue-wise B-factor for all points in the MD simulation, as a measure of global flexibility implemented in cpptraj[66] to identify differences in the conformational diversity between the HA variants. We used PyMOL and ChimeraX to visualize protein structures and electron density maps (PyMOL - The PyMOL Molecular Graphics System, Version 3.0 Schrödinger, LLC). Molecular coordinates for MD models are provided in Supplementary Data 3.

## Cryo-EM sample preparation

HA complexes (CR9114 and LSTa) were prepared by mixing Fab and LSTa respectively with a 3:1 molar ratio of Fab:HA and a 10:1 ratio of LSTa:HA (i.e. 3 Fabs/10 LSTa per HA trimer) and incubated overnight at 4 °C. 0.1% w/v octyl-beta-glucoside detergent was added to the complex to aid in particle tumbling. The Fab was added to further support tumbling. The final concentration of sample was 0.72 mg/mL on the grid. A Vitrobot Mark IV system was used for the preparation of cryoEM grids. The settings were as follows: temperature inside the chamber was 4 °C, humidity was 100%, blotting force was 1, wait time was 0 s, blotting time was varied within a 5.0–5.5 s range. 3 μL of sample was added to plasma cleaned 1.2/1.3 copper Quantifoil 300 mesh grid. The plasma cleaning step was performed in the Solarus 950 plasma system (Gatan) with Ar/O2 gas mix for 25 s. The sample was blotted off for 5 s and the grids were plunge-frozen into liquid-nitrogen-cooled liquid ethane.

## Cryo-EM data collection, processing, and model building

Cryo grids of A/Texas/37/2024 HA – LSTa - CR9114 complexes were imaged at 190,000× nominal magnification using a Falcon 4i camera on a Glacios microscope at 200 kV. Automated image collection was performed using EPU from ThermoFisher. Images were aligned, dose-weighted, and Contrast Transfer Function (CTF)-corrected in the CryoSPARC Live™ software platform, with automated image collection also performed using Smart EPU software (ThermoFisher). Data processing for all three datasets was carried out in CryoSPARC v4.1.2 (https://www.nature.com/articles/nmeth.4169). Blob particle picking was performed on all micrographs with a minimum particle diameter of 100 Å and a maximum of 200 Å. Particles extracted at 512 pixels box size were used to perform 2D classification, which were then used to generate a 3D reference model from ab initio refinement, followed by heterogeneous refinement to obtain one good class that was further non-uniform (NU) heterogeneous refined. Gold-Standard Fourier Shell Correlation (GSFSC) resolution was calculated to be 2.82 Å. As initial coordinates for the HA model we used the predicted Colabfold structure and for CR9114 we used previously published X-ray structures as templates (PDB accession codes: 4FQI and 3UBJ)[25,37]. We docked the models into the cryo-EM density map in UCSF ChimeraX[67]. The structure model was built iteratively with COOT followed by real-space refinement in PHENIX package[68]. The Kabat numbering system was used for mAb and H3 numbering scheme for HA.

## HA modeling

The protein structure 7DEA, from A/duck Northern China/22/2017 (H5N6), was retrieved from the Protein Data Bank (PDB) and visualized using PyMOL (Version 2.6, Schrödinger, LLC). All numbering in this manuscript is H3-numbering, based on Burke and Smith[69].

## Reporting summary

Further information on research design is available in the Nature Portfolio Reporting Summary linked to this article.

## Data availability

All accession identification numbers for H5 sequences used in this study are in Supplementary Data 1. No new sequences were generated in this study. The cryo-EM structure of A/Texas/37/2024 is deposited on Protein Data Bank (PDB) under the accession identification 9DWE. The cryo-EM map and model of CR9114 + LSTa binding rH5 from A/Texas/37/2024 were deposited to the RCSB database with accession numbers EMD-47241 and PDB 9DWE. Previously published structural data used in this manuscript can be found at PDB, including the X-ray structure of A/California/04/2009 HA in complex with LSTa (PDB ID: 3UBJ), A/Vietnam/1203/2004 H5 crystal structure (PDB: 4FQI), and A/duck Northern China/22/2017 H5 structure (PDB: 7DEA). The source data for this study are provided in the Supplementary Information or Source Data files. Source data are provided with this paper.

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

## Acknowledgements

We gratefully acknowledge all data contributors, including the authors and their originating laboratories responsible for obtaining the specimens, and their submitting laboratories for generating the genetic sequence and metadata and sharing via the GISAID Initiative, on which this research is based. We also thank Jian Zheng, Xi Chen, and Nick Berning of Z Biotech for their discussion and suggestions on glycan analyses. This project was funded in part by National Institute of Allergies and Infectious Diseases (NIAID) Centers of Excellence in Influenza Research and Response (CEIRR) grant #75N93021C00045 (J.J.G.) and the Collaborative Influenza Vaccine Innovation Centers (CIVIC) Grant #75N93019C00051 (J.J.G. and A.B.W.). This work was supported in part by The Howard Hughes Medical Institute (HHMI) Emerging Pathogens Initiative (J.J.G.) and the American Heart Association Grant #24PRE1189305 (M.R.G.).

## Author contributions

Conceptualization: M.R.G., W.J., J.J.G.; Methodology: M.R.G., W.J., M.L.F.-Q., A.R., J.H., J.J.G.; Investigation: M.R.G., W.J., M.L.F.-Q., A.R., J.H., J.J.G.; Visualization: M.R.G., W.J., M.L.F.-Q., J.J.G.; Funding acquisition: J.J.G. and A.B.W.; Project administration: J.J.G.; Supervision: J.J.G., J.H. and A.B.W.; Writing – original draft: M.R.G., W.J., J.J.G.; Writing – review & editing: M.R.G., W.J., M.L.F.-Q., A.R., J.H., A.B.W.

## Competing interests

The authors declare no competing interests.
