## [Transparent Peer Review file · Nature Communications]

A single mutation in dairy cow-associated H5N1 viruses increases receptor binding breadth

Corresponding Author: Dr Jenna Guthmiller

Editorial Note: Parts of this peer review file have been redacted as indicated to avoid any copy right infringement.

Version 0:

Reviewer comments:

Reviewer #1

(Remarks to the Author)

Good et al use recombinant HA proteins to examine the receptor preference of recent H5 HA. They observe that the cattle HA has a wider breadth of a2,3 sialic acid binding, specifically the length of the linkages and branching patterns, as well as terminal sugar residues. This is a well done study of timely importance. This reviewer suggests a few minor edits to the figures to enhance the accessibility to the broad readership of Nature Communications.

A few suggestions for improvement:

1. Figure 2: include in a some a2,6 SA data and human seasonal virus (H1N1pdm09 HA) should for reference to compare to previously published data.
2. Figure 4: Inclusion of multiple a2,3 SA like in Fig 2 and the WT Texas cattle strain would be useful
3. Extended data table 1: please aggregate the glycans to a2,6; short a2,3 and long/branched a2,3. To help identify the patterns of banding. The clustered glycans can be shaded for the graph so the reader can easily digest the data.
4. Reference Eisfeld et al 2024 Nature that suggests cattle outbreak virus binds a2.6 SA

Reviewer #3

(Remarks to the Author)

The authors have studied glycan binding by H5 HAs related to recent viruses in cattle. They report that these viruses have a broader recognition of sialylated glycans compared to earlier H5 viral HAs and have data showing that a single mutation in HAs associated with the dairy cow outbreaks may explain the broader receptor recognition due to providing more flexibility in the receptor binding site. The data in general support the conclusions but the following concerns should be addressed.

Line 45: the TCID50 needs a volume specified (eg TCID50 per ml). Line 46:

should be “detectable viral RNA by PCR”

Lines 56-58: There should be some discussion of N-glycolylneuraminic acid in cattle mammary tissue. What is the ratio of NeuAc to NeuGc? The glycan array results in Extended Data Figure 1 show no binding to NeuGc but is there enough NeuAc in mammary tissue to support viral replication?

Lines 118-135 and other places: LSTa is a soluble oligosaccharide that does not itself function as a receptor. It has been used for crystallographic studies simply because it is available in large enough quantities – maybe isolated from bovine milk. In most crystal structures only the terminal sialic acid and maybe two sugars are visible – the rest of the pentasaccharide is unbound and therefore not visible. This appears to be the case in Figure 2 although the sugar units are not labeled. It would be useful to state how many sugar units are visible and which ones are involved in the interactions seen in the MD simulations. Eg line 125 “water mediated hydrogen bond of LSTa with residue E190” is not very helpful.

Which atoms of LSTa and E190?

Where is 199 in Figure 2? It is labeled in Figs 3 and 4 but there is no graphical depiction of the argument that T199I bestows higher flexibility in the receptor binding site.

Line 146: "the amino acid sequence of A/Texas/37/2024 and A/Michigan/90/2015 are identical". Should this be A/Michigan/90/2024?

Line 149: What is the "HA NT sequence"?

Line 189: "Avian influenza viruses are known to have". I think "known" should be replaced with "postulated". Line 235:

For reasons given above, LSTa is not the best mimic of a receptor for further study of "conformations". Extended Data Figure

1: What are NC, PC1-4 and the marker?

Reviewer #4

(Remarks to the Author)

This manuscript presents an interesting study on how variation in avian flu HA can affect receptor breadth, which is potentially very important for understanding how this strain can eventually jump hosts to become a major public health threat to humans. I think that overall, the work is of high quality and obvious importance, and should be published. Here I comment only on the structural modeling and MD analysis, and I have several comments and questions that should be addressed before I would support publication.

First, the structural modeling to generate input models for MD analysis is not sufficiently described. How different are the modeled sequences from similar sequences of known structure? How confident are the AlphaFold models used as input for MD? There should be one or more supplemental figures describing these aspects, since the MD analysis is important for the authors' interpretations of the effects of the T199I mutation. Also lacking is an explanation of how the initial glycan was added to the input model for MD analysis.

Was there a "negative control" for the MD simulations? I think it would be good to include the A/California/04/2009 strain (which was the source of the LSTa glycan coordinates) in the suite of structures for MD - this can provide an anchor of sorts for understanding whether the use of predicted structures versus a crystal structure affected anything.

The authors present a compelling model for how the T199I mutation mediates increased receptor breadth. But the presentation of the data is lacking. In Figure 4C, the dotted line representing a H-bond is not drawn to the N248 amide as noted in the text. Also, how long is this H-bond? It appears quite long. Also, it is not clear from the figure which are the 190 and 220 loop. How does this H-bond stabilize these loops? Similarly, in Figure 4D, can the authors show the B-factors of both the I199 and T199 variants so a reader can directly compare?

Were the B-factors calculated for all time points in the simulations, or only a single time point? Also, if B factors of the loops are as important as the authors say, perhaps a line graph of the B factors of the loops in question would be helpful. It's very difficult to see differences between models when the colors are all shades of blue.

Reviewer #5

(Remarks to the Author)

Version 1:

Reviewer comments:

Reviewer #1

(Remarks to the Author)

All of my comments have been addressed.

Reviewer #3

(Remarks to the Author)

The authors have responded well to the comments on the original submission. They have corrected errors, clarified several points and provided additional data. They have included discussion of three recent publications addressing the receptor specificity of bovine H5N1. One has results showing some binding to an alpha2-6 sialoside while data in two manuscripts in BioRxiv using multiple sialylated glycans have data that agree with the current manuscript that alpha2-6 sialosides do not bind significantly.

I noticed a few typographical errors: Line 195 bonds, not bond

line 215-216 "an analysis SHOWED Asian isolates preferred
Fig 2 legend Line472 conformation, not confirmation Fig 3
legend Line 479 protomer, not promoter

Reviewer #4

(Remarks to the Author)

The authors have addressed all my concerns. Congratulations on a very nice study -

Reviewer #5

(Remarks to the Author)

Reviewer #1 (Remarks to the Author):

Good et al use recombinant HA proteins to examine the receptor preference of recent H5 HA. They observe that the cattle HA has a wider breadth of α 2,3 sialic acid binding, specifically the length of the linkages and branching patterns, as well as terminal sugar residues. This is a well done study of timely importance. This reviewer suggests a few minor edits to the figures to enhance the accessibility to the broad readership of Nature Communications.

We greatly appreciate the reviewer's nice comments about our manuscript and strongly believe their edits have strengthened the manuscript.

A few suggestions for improvement:

1. Figure 2: include in a some α 2,6 SA data and human seasonal virus (H1N1pdm09 HA) should for reference to compare to previously published data.

We have now updated Extended Data Figure 1 to show clearly which glycans are α 2,6 or α 2,3, which clearly show high specificity to α 2,3 Neu5Ac. We have added references that used this same microarray with H1N1, which showed high specificity to α 2,6 sialic acids (Lines 107-108). In our hands, we also see high specificity of A/California/4/2009 recombinant H1 binding to select α 2,6 Neu5Ac sialic acids (see below). As these data were generated to serve as an important control for another publication on H1 evolution within the receptor binding site, we have opted to not include these data in this manuscript to avoid data duplication.

[redacted]

We have now added this to Fig. 5, formerly Fig. 4, to improve clarity.

2. Extended data table 1: please aggregate the glycans to α 2,6; short α 2,3 and long/branched α 2,3. To help identify the patterns of banding. The clustered glycans can be shaded for the graph so the reader can easily digest the data.

We believe this comment is for Extended Data Figure 1, not Extended Data Table 1. We have now included shading to help discern these different linkages and to separate sialylated LacNAcs from sialyl Lewis X structures, as well as Neu5Ac and Neu5Gc glycans. However, we are unable to discern based on branch length as sialylated

branches were the same length, whereas the second arm may be of varying lengths and lack sialylation. We believe the new shading schema now helps the reader understand binding profiles more clearly. We have also added text at lines 105-107 to clarify that a single LacNAc or sialyl lewis X moiety is sialylated and there are no repeats/long branches. Moreover, we have included in the study limitation (lines 278-282) that the array does not have poly-LacNAc structures, which are known to be the preferred receptor for recent H3N2 viruses.

3. Reference Eisfeld et al 2024 Nature that suggests cattle outbreak virus binds a2.6 SA

We have now added to the discussion this manuscript and how it is inconsistent with our findings (Lines 248-258). Moreover, we have added a few sentences about two pre-prints that have subsequently come out that also show dairy cow-associated H5N1 viruses are highly specific to a2,3 sialic acids. Importantly, these additional studies use additional widely accepted approaches to study receptor binding specificity, including microarrays and ELISAs. Nicely, one of these studies (Santos et al.) uses recombinant HA, similar to our manuscript, and the other study (Chopra et al.) uses intact virus, more akin to the Eisfeld et al. study. Therefore, we are confident in the conclusion that H5 is specific to a2,3 sialic acids.

Reviewer #3 (Remarks to the Author):

The authors have studied glycan binding by H5 HAs related to recent viruses in cattle. They report that these viruses have a broader recognition of sialylated glycans compared to earlier H5 viral HAs and have data showing that a single mutation in HAs associated with the dairy cow outbreaks may explain the broader receptor recognition due to providing more flexibility in the receptor binding site. The data in general support the conclusions but the following concerns should be addressed.

We appreciate the reviewer's positive feedback and believe we have addressed all of their concerns.

Line 45: the TCID50 needs a volume specified (eg TCID50 per ml).

We appreciate the reviewer catching this. We have now clarified that it is TCID50 per ml.

Line 46: should be "detectable viral RNA by PCR"

We have corrected this, now at line 46.

Lines 56-58: There should be some discussion of N-glycolylneuraminic acid in cattle mammary tissue. What is the ratio of NeuAc to NeuGc? The glycan array results in Extended Data Figure 1 show no binding to NeuGc but is there enough NeuAc in mammary tissue to support viral replication?

The reviewer raises an important point that was not addressed in the prior iteration of the manuscript. We have now added text at lines 101-104 about the nature of Neu5Gc and the absence of binding to Neu5Gc. In discussion, we have further discussed the role of Neu5Gc, including that it is known to be present in the mammary tissue. Moreover, we have added a reference highlighting that Neu5Ac is 10 to 100 times more abundant in cow's milk relative to Neu5Gc (lines 223-224).

Since we found no binding of H5 to Neu5Gc and a recent study found H5 binds robustly to cow mammary tissue, we deduce that α 2,3 Neu5Ac serves as the receptor for H5N1 within dairy cows. Moreover, a recent study that directly inoculated cow udders with H5N1 saw a robust infection of this tissue, which further supports that Neu5Ac within the cow mammary tissue likely facilitates infection (lines 224-228).

Lines 118-135 and other places: LSTa is a soluble oligosaccharide that does not itself function as a receptor. It has been used for crystallographic studies simply because it is available in large enough quantities – maybe isolated from bovine milk. In most crystal structures only the terminal sialic acid and maybe two sugars are visible – the rest of the pentasaccharide is unbound and therefore not visible. This appears to be the case in Figure 2 although the sugar units are not labeled. It would be useful to state how many sugar units are visible and which ones are involved in the interactions seen in the MD simulations. Eg line 125 “water mediated hydrogen bond of LSTa with residue E190” is not very helpful. Which atoms of LSTa and E190?

We thank the reviewer for this comment. We have added Extended Data Figure 2 which highlights the structure of LSTa. Moreover, we have now added a cryo-EM structure of A/Texas/37/2024 HA binding to LSTa and have included a table of contacts (Figure 3C), which indicates only binding contacts with Neu5Ac and galactose. Similarly, resolution for LSTa could only be made for the Neu5Ac and galactose-2, due to the flexible nature of glycans. Since MD modeling is based on known structures and structures rarely show the additional sugars, it is not feasible to model these interactions.

Where is 199 in Figure 2? It is labeled in Figs 3 and 4 but there is no graphical depiction of the argument that T199I bestows higher flexibility in the receptor binding site.

Since the T199I mutation is not discussed until Figure 4 (formerly figure 3), we have opted to not label it in Figure 2. We do show in Figure 2D and Figure 4D (A/Texas/37/2024 with T199) that there are differences in flexibility by B-factor. To help illustrate this further, we have now added Figure 5E, which shows the B-factor across the 130-loop, 190-helix, and 220-loop of A/Colorado/18/2022, WT A/Texas/37/2024, and A/Texas/37/2024 T199. We believe that this figure now nicely summarizes these differences in flexibility across these three strains.

Line 146: “the amino acid sequence of A/Texas/37/2024 and A/Michigan/90/2015 are identical”. Should this be A/Michigan/90/2024?

Thank you for noticing this! We have corrected this.

Line 149: What is the “HA NT sequence”?

We have now added this as Extended Data Figure 4.

Line 189: “Avian influenza viruses are known to have”. I think “known” should be replaced with “postulated”.

We have changed this wording.

Line 235: For reasons given above, LSTa is not the best mimic of a receptor for further study of “conformations”.

We have now added a cryo-EM structure of A/Texas/37/2024 to the manuscript (Fig. 3 and Extended Fig. 3). The line referred to is now removed, as now we have a deeper and experimentally validated structure of how rH5 binds to LSTa. While LSTa is not the most physiologically relevant sialic acid to use to study HA-sialic acid interactions, it is a useful model sialoside for understanding contacts within the HA RBS with sialic acid. We have added this point to the “limitation of study” (line 282-286).

Extended Data Figure 1: What are NC, PC1-4 and the marker?

These definitions have been added to the figure legend for Extended Data Fig. 1.

Reviewer #4 (Remarks to the Author):

This manuscript presents an interesting study on how variation in avian flu HA can affect receptor breadth, which is potentially very important for understanding how this strain can eventually jump hosts to become a major public health threat to humans. I think that overall, the work is of high quality and obvious importance, and should be published. Here I comment only on the structural modeling and MD analysis, and I have several comments and questions that should be addressed before I would support publication.

We greatly appreciate the reviewer’s feedback and believe these additions and data clarifications strengthen the manuscript.

First, the structural modeling to generate input models for MD analysis is not sufficiently described. How different are the modeled sequences from similar sequences of known structure? How confident are the AlphaFold models used as input for MD? There should be one or more supplemental figures describing

these aspects, since the MD analysis is important for the authors' interpretations of the effects of the T199I mutation. Also lacking is an explanation of how the initial glycan was added to the input model for MD analysis.

We appreciate the reviewer's comment and added additional information in the methods section of the manuscript. The AlphaFold models of H5, which have been used as input for MD simulations revealed high levels of confidence with an overall PLDDT score of over 90%. The initial glycans have been added using GlycoShape and by using previously available information on H5 glycosylation. To address the reliability of the AlphaFold predictions, we got a 2.85 Å-resolution cryo-EM structure of hemagglutinin A/Texas/37/2024 binding LSTa, further emphasizing the high accuracy of the AlphaFold prediction (new Fig. 3). We have overlaid the cryo-EM structure with the most probable representative MD model, which revealed nearly identical structures (Extended Data Fig. 3D).

Was there a "negative control" for the MD simulations? I think it would be good to include the A/California/04/2009 strain (which was the source of the LSTa glycan coordinates) in the suite of structures for MD - this can provide an anchor of sorts for understanding whether the use of predicted structures versus a crystal structure affected anything.

We thank the reviewer for this comment. To address this comment, as mentioned above we determined the structure of A/Texas/37/2024 in complex with LSTa and the antibody CR9114 to improve tumbling to ensure the reliability of our predictions. We provided an overlay of the cryo-EM structure with our initial model in Extended Data Fig. 4D, which supports the model accuracy.

The authors present a compelling model for how the T199I mutation mediates increased receptor breadth. But the presentation of the data is lacking. In Figure 4C, the dotted line representing a H-bond is not drawn to the N248 amide as noted in the text. Also, how long is this H-bond? It appears quite long. Also, it is not clear from the figure which are the 190 and 220 loop. How does this H-bond stabilize these loops? Similarly, in Figure 4D, can the authors show the B-factors of both the I199 and T199 variants so a reader can directly compare?

We agree with the reviewer. In Figure 4C, we have updated the figure showing all contact residues of T199 that are involved in forming direct or water-bridged interactions more than 30% of the time (Fig. 5C and lines 191-194). An interaction/water-mediate interaction was defined based on a 3.5 Å heavy atom distance-cutoff criteria. We added these details to the methods. In total, T199 and N248 interact 80% with each other (including water bridges). In addition, T199 interacts with R212 in the same and in the neighboring protomer via a water bridge approximately 30% of the simulation, further emphasizing the additional stabilizing role of T199. We included these data in the text and provided a new visual of T199 and its interacting partners. This image does not show the 190-helix and 200-loop, so we have decided to remove this text at this point in the manuscript.

We have now added Fig. 5E (as suggested in the next comment), which shows the B-factor of each residue in the 130- and 200-loops and the 190-helix for the three HAs that MD models were generated. These data nicely show how the single mutation at T199I greatly influences flexibility in these three regions that are critical for binding to sialic acid.

Were the B-factors calculated for all time points in the simulations, or only a single time point? Also, if B factors of the loops are as important as the authors say, perhaps a line graph of the B factors of the loops in question would be helpful. It's very difficult to see differences between models when the colors are all shades of blue.

We appreciate the comment from the reviewer and clarified the methods section. The B-factors were calculated for all points in the MD simulation and we provided plots showing line graphs of the B-factors for the loops/helix in question (Fig. 5E).

Reviewer #5 (Remarks to the Author):

Thank you for your contribution and we hope this was a great learning experience!

Response to Reviewer (Reviewer in bold, our response in normal typeface)

Reviewer #1:

All of my comments have been addressed.

No further edits necessary.

Reviewer #3:

The authors have responded well to the comments on the original submission. They have corrected errors, clarified several points and provided additional data. They have included discussion of three recent publications addressing the receptor specificity of bovine H5N1. One has results showing some binding to an alpha2-6 sialoside while data in two manuscripts in BioRxiv using multiple sialylated glycans have data that agree with the current manuscript that alpha2-6 sialosides do not bind significantly.

I noticed a few typographical errors:

Line 195 bonds, not bond

line 215-216 "an analysis SHOWED Asian isolates.... preferred

Fig 2 legend Line472 conformation, not confirmation

Fig 3 legend Line 479 protomer, not promoter

We appreciate the reviewer identifying these errors. We have corrected each one within the manuscript and figure legends.

Reviewer #4:

The authors have addressed all my concerns. Congratulations on a very nice study -

No further edits necessary.